# FSR Systems for Detection of Air Objects Using Cosmic Radio Emissions

**DOI:** 10.3390/s21020465

**Published:** 2021-01-11

**Authors:** Hristo Kabakchiev, Vera Behar, Ivan Garvanov, Dorina Kabakchieva, Avgust Kabakchiev, Hermann Rohling

**Affiliations:** 1Faculty of Mathematics and Informatics, Sofia University, 1164 Sofia, Bulgaria; 2Mathematical Methods for Sensor Data Processing Department, Institute of Information and Communication Technologies, 1113 Sofia, Bulgaria; behar@bas.bg; 3Information Systems and Technologies Department, University of Library Studies and Information Technologies, 1784 Sofia, Bulgaria; i.garvanov@unibit.bg; 4Faculty of Applied Informatics and Statistics, University of National and World Economy, 1700 Sofia, Bulgaria; dkabakchieva@unwe.bg; 5BULATCA, 1540 Sofia, Bulgaria; av_kab@abv.bg; 6Institute of Communications, Technical University Hamburg-Harburg, 21073 Hamburg, Germany; rohling@tu-harburg.de

**Keywords:** Fraunhofer diffraction, forward scatter effect, object detection

## Abstract

The paper analyses the possibility of Forward Scatter Radar (FSR) systems to detect airplanes using cosmic emission from pulsars and planets (pulsar, Sun, Moon). A suboptimal multichannel algorithm for joint detection and evaluation of the parameters of the forward scattering signal created by an airplane (duration and velocity) is proposed, with preliminary compensation of the powerful direct signal emitted by cosmic sources (pulsar, Sun and Moon). The expressions for calculation of the Signal-to-Noise Ratio (SNR) at the input of the detector and the compensator are obtained. The detection characteristics are also obtained, and the requirements for the suppression coefficient of the compensator are evaluated. A methodology for calculating the maximum distance for detecting an aircraft using a described algorithm is proposed. The obtained results show that due to the Forward Scatter (FS) effect, there is the theoretical possibility to detect airplanes at close ranges by FSRs, which use very weak signals from cosmic sources.

## 1. Introduction

In the early 1950s, the United States entered into service the AN/SAN-25 passive radar, designed to determine the angular coordinates of extraterrestrial radio sources and used to navigate aircrafts and ships. The first passive radars used radiation from the Sun and then from the Moon. Attempts have also been made to use the radio emission of pulsars in passive radars (for example, Cassiopeia) [1,2,3]. The main problem with the widespread use of passive radars was the creation of highly sensitive receivers to receive very weak signals from space. The advent of artificial space radio transmitters as well as the 5G system has led to the development of other radio navigation and communication systems that we now use. A number of variants of passive radars have been proposed, which use signals from various modern communications sources (Secondary Application of Wireless Technology). Forward Scatter Radar (FSR) is a specific type of passive radar, named as a radio electronic barrier. Recently, barrier FSR have become very relevant for the protection of various types of space—land, sea, air and space. The appearance of the FS effect at the crossing of the barrier allows to successfully detect small sizes targets, including those with the Stealth technology. For this reason, FSR systems for ground, sea, air and space purposes are currently being developed and sold, combined with traditional radar systems. However, FSR systems have one major drawback—there is no resolution in distance. The target is detected, but not the distance to it. For this reason, they are used in combination with traditional radars. The principle of operation of such radars is based on the theory of Fraunhofer diffraction of electromagnetic waves and the Babinet principle [4,5,6,7,8]. According to diffraction theory, when an airplane is near the baseline (a bistatic angle is approximately 180°), the object generates a strong shadow field repeatedly increasing the shadow Radar Cross Section (RCS) of the object [9,10,11]. The FSR systems have a small area of coverage (in the vicinity of the baseline between the transmitter and the receiver) and a small delay of the scattered signal relative to the direct transmitter signal, which directly arrives at the receiver input. The short delay time of the signal from the target relative to the direct signal does not make it possible to separate these signals in the time domain, which makes it difficult to compensate for the direct signal. A small delay of the useful signal compared to the direct signal allows the creation of FSR systems with the external coherence, which can measure the speed of the aircraft. The phenomena of forward scattering based on the theory of Fraunhofer diffraction (far zone of diffraction) when the target is located very far from the receiver and the transmitter. The cosmic transmitters are located very far from the target. However, the Cassini’s theory proposes an equation of the distances “target-receiver” and “target-source” in bistatic radar systems. When the distance between the receiver and the space source is very large (millions of km), then according to the Cassini equation, two ellipses are formed, where detection is possible.

As for the distance from the receiver to the target, this distance, according to the theory of Fraunhofer diffraction should be more than Q/λ. This is the distance, from which begins the far zone of diffraction. That is why all the calculations are done for distances between the receiver and the target, which more than Q/λ.

In the last ten years, there has been a particular interest in these bistatic FSR systems, since they practically allow to observe such targets (Stealth technology), which are invisible by conventional radar systems [12,13,14]. The practical use of FSR systems goes together with bistatic radar systems. The purpose is to take the advantages of the Forward Scatter effect on the one hand and to offset its disadvantages, in determination of the parameters of moving targets. Another possible way is the use of natural alternative signal sources in the FSR systems for detecting airplanes.

Such natural sources of radio signals can be cosmic bodies emitting radio signals, which can be received by radio telescopes on the earth [1,2,13,14]. Although these cosmic radio signals are very weak, they can be efficiently processed in radio telescope receivers, which would not be possible with traditional radars [7]. The standard algorithm for processing of pulsar signals implemented in the radio telescopes usually includes signal accumulation. The signal processing algorithm for detection of pulsar signals that use the filtering approach is described in [13]. The other cyclostationary approach is presented in [14,15]. 

The topology of an FSR system that exploits cosmic sources of radio signals is presented in Figure 1.

When crossing the barrier between the space emitter and the FSR at a right angle from an airplane, a new signal is induced and is reflected forward by the aircraft. The energy of this signal is proportional to the area of the aircraft and this allows very small flying objects or objects using stealth technology to be detected. According to Umtsev [12], the shadow radiation does not depend on the whole shape or material of the shadowing object and is completely determined by the size and the geometry of the projected contour of the object, perpendicular to the observation axis. This approximation can only be valid if the target is considered as absolute black body, which is valid for the general radar target, i.e., vehicles, people and animals, which resolves the biggest advantage of FSR—the received signal is not acted by the scattering coefficients of the target and thus stealth coating or geometry do not reduce the FSRs. In the shadow area, the area that is close to the receiver (up to 1 km), the received signal from the plane is much stronger than the direct signal and that is why the second one is suppressed and forms a forward scattering signal of the plane in the receiver. The length of the signal is proportional to the geometry of the aircraft. This can be seen in the photo in Figure 2 kindly provided by the Rozhen Observatory, Bulgaria. The photo shows the shadow of an airplane against the lunar background in an FSR system “Moon—camera”. Aircraft detection in the FSR is usually performed in the far zone of Fraunhofer diffraction (5–40 km), where the FS signal from the aircraft is much weaker than the direct one, which requires a compensation of the direct one. In FSR with a space emitter, direct signals are much weaker than receiver noise. Because of that correlation receiving of space signals must be performed first in order to estimate their parameters. After that, the compensation of the estimated direct signal is performed. The detection of a signal from the target is performed by making a correlation between the estimation of the signal from the aircraft (supporting signal) and the additive sum of the signal of the target on the background of the receiver noise. The received result from the correlation is compared with a pre-selected threshold.

The goal of this paper is to present and describe the suitable structure of an algorithm for jointly detecting and evaluating the parameters of the forward scattering signals from airplanes in FSR, which uses radio signals from the natural cosmic sources—pulsars, Sun, Moon. Specifically, the article proposes a new modified multichannel algorithm for joint detection and estimation of the parameters of a quasi-deterministic forward scattering signal from an airplane, with preliminary compensation of a powerful direct quasi-deterministic signal from the Sun, Moon and pulsar before detecting a forward scattering signal against the background of a Gaussian receiver noise.

The estimated parameters of quasi-deterministic signals (direct and shadow) are a priori unknown. These parameters are the duration of the signals that depend on the contact time of the receiving antenna beam with the airplane beam and also the airplane speed. The article does not discuss the method and the algorithm of the direct signal estimation on the background white noise and its compensation. The estimated useful signal parameters are non-random variables and do not change over time, and that is why the Kalman filter is not used. For detection and estimation, the same multichannel algorithms based on the estimation of the maximum likelihood are used.

In this article, the probabilistic characteristics for airplane detection are also obtained depending on the probability of a false alarm and the energy Signal-to-Noise Ratio (SNR) at the signal detector input. The energy SNR at the input of the signal detector is calculated depending on the cosmic transmitter parameters, the receiver parameters, the parameters of airplanes that are crossing the baseline “transmitter-receiver” at the right angle, and also on airplane distances.

The article is organized into the following sections: Introduction, Operating Principle of FSR with Cosmic Radio Emission, Signal Models in FSR Systems with Cosmic Radio Emission, Structure of the Multichannel Algorithm for Detecting the Forward Scattering Signal from an Airplane, Power Budget of FSR Systems, Probability Characteristics, Results, Conclusion, and References.

## 2. Operating Principle of FSR with Cosmic Radio Emission

As is known, only some parts of electromagnetic waves of the radio range reach the surface of the earth. There is a wide window in radio frequencies, which allows seeing the cosmic space at radio frequencies. The FSR systems that exploit natural sources of radio signals (pulsar, Sun, Moon) give the possibility for detection of moving targets (airplanes). In the absence of an airplane on the baseline “transmitter-receiver”, a direct unobstructed signal $U_{direct}$ arrives at the receiver input from the cosmic source. However, in a case when any airplane crosses the baseline “transmitter-receiver”, the direct signal $U_{direct}$ is blocked by the airplane. The input signal $U_{RX}$ of the receiver can be represented as a sum of the direct unobstructed signal $U_{direct}$ and the blocking forward scattering signal $U_{shadow}$ [12]:(1)$$U_{RX}=U_{direct}+U_{shadow}$$

At the principle of Babinet, the direct incident radiation $U_{direct}$ and shadow radiation $U_{shadow}$, created by an opaque airplane, have opposite phases [7]. In the theory of diffraction [12], the mathematical description of the shadow field created by an airplane is based on the Kirchhoff–Fresnel integral [7,12]:(2)$$U_{shadow}=\frac{j}{\lambda }\iint_{Q}^{}U_{direct}\frac{\exp\left(-jkR\right)}{R}dq$$

In Equation (2), Q is a silhouette area of the airplane, dq is a small part of the area of the airplane silhouette, *R* is a distance from that part to the receiver, and λ is the wavelength of emission. As can be seen, a shadow field does not depend on the shape or material of the shadowing airplane but is entirely determined by the geometry of the airplane silhouette. According to the theory of Fraunhofer diffraction [12], in the far diffraction zone, when the airplane with the silhouette area *Q* is located at distance *R* from the receiver and the distance *R* satisfies the inequality *R > Q/λ* (far diffraction zone), it can be assumed that: (i)—the direct field $U_{direct}$ from the transmitter is distributed evenly over the area of the airplane silhouette; (ii)—the value of the distance R almost does not change with a change in the position of dq on the area of the airplane silhouette. In that case after integration Equation (2) takes the form:(3)$$U_{shadow}=\frac{jexp\left(-jkR\right)}{\lambda R}U_{direct}Q$$

As follows from Equation (3) the envelope of the forward scattering signal can be expressed by the following equation:(4)$$\left|U_{shadow}\right|=E_{shadow}=E_{direct}Q/\left(\lambda R\right)$$

Because the direct incident radiation and shadow radiation from an airplane have opposite phases, the envelope on the input signal of the receiver is determined as:(5)$$E_{RX}=E_{direct}-E_{shadow}$$

Therefore, the forward scattering signal $E_{shadow}\;$can be considered as a reduction (leakage) of a direct signal $\;E_{direct}$, received in an FSR system. Hence, will have the following inequality:(6)$$E_{shadow}\leq E_{direct}$$

Taking into account Equations (4) and (6), it can be concluded that the calculation of SNR at the detector input can be performed only for distances (R) to airplanes satisfying the following inequality:(7)R≥Q/λ

This means that the detection of airplanes in an FSR system can be performed only for distances (R) satisfying the inequality Equation (7). In an FSR system, the gain of the signal (q) at the output of the radio receiver due to the integration of the received signal can be expressed as:(8)$$q=\sqrt{\Delta ft_{cont}}$$

In Equation (8), Δf is the frequency bandwidth of the receiver and $t_{cont}$ is the airplane beam contact time with the beam of the receiving antenna. The contact time that is the duration of the received signals is calculated as:(9)tcont=min{λl,λd}R/Vlat
where λd is the angular beamwidth of the receiving antenna, λl is the angular beam width of the shadow radiation from the airplane, $V_{lat}$ is the airplane lateral velocity, λ is the wavelength, and R is the distance to the airplane. In the case of pulsed FSR, the contact time determines the length of a series of pulses re-emitted from the airplane to the receiver, the number of which is:(10)$$N_{p}=t_{cont}/P$$

In Equation (10), P is the repetition period of pulsar pulses. It is known that the energy SNR at the input of a signal detector realized in the LF domain (e.g., CFAR) depends on both the energy SNR at the receiver input and the parameter q of the receiver [16,17,18]. 

## 3. Signal Models in FSR Systems with Cosmic Radio Emission

It is known that when the airplane is located near the baseline between the receiver and the source is observed the Forward Scatter (FS) effect at which the receiver of an FSR system receives at the same time the radio signal, emitted by the cosmic source (direct signal), and the shadow, created by the airplane (target signal), on the background of the receiver noise [7]. The input signal of the detector is:(11)$$y\left(t\right)=s_{direct}\left(t\right)+s_{target}\left(t\right)+N\left(0,P_{n}\right)$$

The signal components in Equation (11) have a different form in the case of pulsar FSR and solar/lunar FSR systems, where $P_{n}\;$is the power of the receiver noise.

*Pulsar FSR.* In the case of a pulsar FSR system, the receiver receives an additive sum of the pulse sequence, emitted by the pulsar (direct signal) and the pulse sequence, created by the airplane (target forward scattering signal), on the background of the receiver noise. At the input of the pulsar FSR system, the direct signal from a pulsar, mathematically can be represented as a convolution between the pulsar pulse shape function (pulse profile) p(t) and the Shah periodic function:(12)$$s_{direct}\left(t\right)=\sqrt{P_{imp}}\;p\left(t\right)\cos\left(2\pi f_{0}t\right)\sum_{k=-\infty }^{N_{P}}\delta \left(t-kP\right)\;0\leq t\leq t_{cont}$$

In Equation (12), $f_{0}$ is the central frequency of pulsar emission, P is the repetition period of pulsar pulses and $N_{P}$ is the number of pulses received during the contact time, $P_{imp}$ is the power of the single impulse. The parameter $P_{imp}$ will be explained in Section 4, Equation (26). The pulsar pulse profile p(t) is specific for each pulsar according to the European Pulsar Network (EPN) [10]. We assume that the pulse sequence of the airplane forward scattering signal does not change the pulse form and the repetition period. The pulses of the airplane forward scattering signal are shifted in frequency (Doppler shift) depending on the airplane velocity. The amplitude of the airplane forward scattering signal sufficiently depends on the distance to the airplane and his shadow Radar Cross Section (RCS). According to the Babine principle we assume that the airplane forward scattering signal has the same structure as a direct signal, it can be represented as a power-attenuated and frequency-shifted copy of the direct signal:(13)$$s_{target}\left(t\right)=\sqrt{bP_{imp}}p\left(t\right)\cos\left[2\pi \left(f_{0}+\frac{2V}{\lambda }\right)t\right]\sum_{k=-\infty }^{N_{p}}\delta \left(t-kP\right)\;0\leq t\leq t_{cont}$$
where V is the radial velocity of an airplane, λ is the wavelength of radio emission and b is the coefficient of attenuation, i.e., b≪1 because the forward scattering signal from an airplane is much weaker than the direct signal [7].

*Solar/lunar FSR.* In a solar/lunar FSR system, the direct signal can be represented as a band-limited signal with the power $P_{solar}$ for the Sun ($P_{lunar}$ for the Moon) and the envelope form A(t).
(14)$$s_{direct}\left(t\right)=\sqrt{P_{solar}}A\left(t\right)cos\left(2\pi f_{0}t\right),\;0\leq t\leq t_{cont}$$

In analogy, the airplane forward scattering signal can be represented as a frequency-shifted version of the narrow-band signal that is weaker than the direct signal:(15)$$s_{target}\left(t\right)=c\sqrt{P_{solar}}A\left(t\right)cos\left(2\pi (f_{0}+\frac{2V}{\lambda }\right)t),\;0\leq t\leq t_{cont}$$
where c is the factor of attenuation, c≪1. These models of quasi-deterministic signals with the deterministic waveform A(t) and unknown parameters (V and $t_{cont}$) are chosen here for convenience. The parameters $P_{solar}$ and $P_{lunar}\;$will be explained in Section 4, Equation (36). These models lack such random parameters of the quasi-deterministic signal as a random amplitude or phase, traditionally used in radars.

## 4. Structure of the Multichannel Algorithm for Detecting the Forward Scattering Signal from an Airplane

The task of detecting an aircraft in FSR using radio emission from cosmic sources (Sun, Moon, pulsar) can be formulated in a statistical sense as the task of joint detection and evaluation of parameters (Doppler shift, duration) of a shadow radio signal from the aircraft against the background of a strong direct radio signal from the cosmic source and limited Gaussian noise of the receiver. White Gaussian noise is commonly used as a receiver noise model in radio astronomy. This is because White Gaussian noise very well approximates the own noises of the radio receiver and the noises received from space. The noises are normalized and their spectrum width is usually much larger than the bandwidth of the receiver.

The goal of that paper is to synthesize a new modified algorithm, suitable for solving this statistical task, basing on a well-known statistical algorithm for joint detection and evaluation of the parameters of the useful signal against the background of strong interference and white Gaussian noise. This new modified algorithm must take into account the following limitations on signals: (i)—the direct signal and the forward scattering signal from the airplane have the same structure and are described as quasi-deterministic signals; (ii)—the direct signal is much more powerful than the weak forward scattering signal from the airplane; (iii)—the two radio signals, direct and shadow, have unknown parameters; (iv)—the direct signal is considered as a strong interference compared to the forward scattering signal from the airplane.

Different statistical algorithms for detecting and evaluating signal parameters, as well as algorithms for suppressing, selecting or compensating the interference have been described in the literature [4,7]. The statistical task for the joint detection and evaluation of the parameters of two quasi-deterministic signals against the background of a white Gaussian noise is known and has been solved long ago. It is solved by using the classical approaches for signal selection with subsequent joint detection and evaluation of the detected useful signal [2,7].

In the article it is assumed that the radio signals, received in FSR from the cosmic source of radio signals and the airplane, are quasi-deterministic signals with unknown parameters: duration of the direct and the forward scattering signal from the airplane (the contact time—$t_{cont}$) and the airplane velocity (V).

It is also assumed that at the input of the receiver of FSR arrives from the additive mixture of three signals:(16)$$y\left(t,t_{cont},V\right)=s_{target}\left(t,t_{cont},V\right)+s_{direct}\left(t,t_{cont}\right)+n\left(t\right),\;0\leq t\leq t_{cont}$$

The practical task of detecting an airplane in the FSR can be formulated as a statistical task of detecting a forward scattering signal from an airplane ($s_{target}\left(t\right)$) against the background of strong interference ($s_{target}\left(t\right)$) and receiver noise (n(t)) at unknown parameters $t_{cont}$ and V.

The estimated parameters in the task are $t_{cont}$ and V. To simplify the search for a suitable algorithm, the task of detecting the forward scattering signal against the background of a strong direct signal can be divided into two separate tasks. To solve the first and traditional radar problems, it is necessary to find an algorithm for joint detection and estimation of the parameters $t_{cont}$ and V of the useful forward scattering signal $s_{target}\left(t,t_{cont},V\right)$ under the assumption that preliminary compensation (removal) of thedirect signal is performed. The second non-traditional task for radars is the compensation of the direct signal $s_{direct}\left(t,t_{cont}\right)$, i.e., subtraction of the direct signal with non-random and unknown duration from the input signal of the receiver $y\left(t,t_{cont},V\right)$.

The structure of a multichannel algorithm for joint detection and estimation of one or two parameters of a quasi-deterministic signal against the background of the receiver noise can be used as the basis for an algorithm for detecting a forward scattering signal from an aircraft [8]. Since the estimated parameters are non-random variables, the criterion of the maximum likelihood ratio can be used in the synthesis of the optimal detector. In the case of a single estimated non-random parameter (e.g., $t_{cont}$), the optimal decision rule for signal detection is to compare the maximum of the log likelihood ratio with a pre-selected threshold. The signal detection algorithm in this system of joint signal detection and non-random parameter estimation is optimal according to the criterion of the maximum likelihood ratio. The algorithm for estimating the parameter is also optimal, as it forms an estimate of the maximum likelihood.

It is known that in the case of estimating two non-random parameters $\left(t_{cont},\;V\right)$ this algorithm for joint detection and estimation retains its structure and type of the decision rule. Since in our case the direct and forward scattering signals are considered as quasi-deterministic signals with unknown parameters $\left(t_{cont},\;V\right)$, the statistics (the logarithm of the likelihood ratio) in the multichannel algorithm for joint detection and estimation of these parameters in each channel (i, j), where i=1…M and j=1…N, has the form [2,7]:(17)$$Z_{i,j,T}\left({\hat{t}}_{cont,i},{\hat{V}}_{j}\right)=\frac{2}{N_{0}}\int_{0}^{T}s_{target}(t,{\hat{t}}_{cont,i},{\hat{V}}_{j})y\left(t\right)dt-\frac{2}{N_{0}}\int_{0}^{T}[s_{target}{\left(t,{\hat{t}}_{cont,i},{\hat{V}}_{j}\right]}^{2}dt$$
where $T={\hat{t}}_{cont,i}$. The parameters ${\hat{t}}_{cont,i}$ and ${\hat{V}}_{j}$ are the estimates of the contact time $t_{cont}$ and the speed V of the aircraft in the (i, j) channel. The parameter $N_{0}\;$is the spectral density of the white noise of the receiver. In each channel the logarithm of the likelihood ratio is formed for fixed values of a pair of estimated parameters $\left(t_{cont,i},\;V_{j}\right)$, taken from the set of their possible values. The number of channels is determined by the accuracy of the parameter estimates. The values of a pair of parameters $\left(t_{cont},\;V\right)$ in the channel, where the logarithm of the likelihood ratio has a maximum value and exceeds the detection threshold, are assumed as the parameter estimates, i.e., ${\hat{t}}_{cont}$ и $\hat{V}$. The described algorithm for joint detection and evaluation of the parameters of the forward scattering signal is without compensation of the direct signal.

According to [8], in the case when the direct signal is compensated (removed), the statistics (logarithm of the likelihood ratio) for our quasi-deterministic signals with unknown parameters $\left(t_{cont},\;V\right)$, in the multichannel algorithm for detection and estimation of the useful signal parameters, in each channel (i, j), where i=1…M and j=1…N, if the determined quantities $\frac{2}{N_{0}}\int_{0}^{T}[s_{target}{\left(t,{\hat{t}}_{cont,i},{\hat{V}}_{j}\right]}^{2}dt$, are included in the detection threshold it takes the form: (18)$$Z_{i,j,T}\left({\hat{t}}_{cont,i},{\hat{V}}_{j}\right)=\frac{2}{N_{0}}\int_{0}^{T}s_{target}(t,{\hat{t}}_{cont,i},{\hat{V}}_{j})y_{0,i}\left(t\right)dt$$
(19)$$y_{0,i}\left(t\right)=y\left(t\right)-s_{direct}\left(t,{\hat{t}}_{cont,i}\right)\;\mathrm{and}\;T={\hat{t}}_{cont,i}$$

The signal $y_{0,i}\left(t\right)$ in Equation (19) is a signal at the input of the detector after compensation (removal) of the direct signal from the input signal y(t). It is the difference between the input signal y(t) and the estimate of the direct signal$\;s_{direct}\left(t,{\hat{t}}_{cont,i}\right)$, obtained in the absence of the forward scattering signal.

The detection and evaluation of the parameters of the useful (informative) signal is performed with the same algorithm for the maximum likelihood Equation (18). This is because maximizing the likelihood function to find the estimate is equivalent to maximizing the conditional likelihood ratio (or logarithm). This allows for the use of algorithms for optimal signal detection in the synthesis of optimal estimators of signal parameters.

Taking into account Equations (18) and (19), the proposed general structure of an algorithm for joint compensation, detection, and estimation of the parameters of the quasi-deterministic forward scattering signal from an airplane against the background of a strong direct signal from the cosmic source of radio signals and the receiver noise will be as follows (Figure 3).

According to Figure 3, in each channel (i, j), where i=1…M and j=1…N, the multichannel algorithm successively compensates the direct signal and forms the logarithm of the likelihood ratio $Z_{T}$.

The number of channels is determined by the accuracy of estimating the parameters, where M is the number of $t_{cont}$ channels and N is the number of V–channels. The deciding rule selects that channel, where the logarithm of the likelihood ratio has a maximum value, and compares the value of the logarithm of the likelihood ratio with the predetermined threshold (H). If the threshold is exceeded, the forward scattering signal from the aircraft is considered detected and the number of the respective channel determines the estimates of the parameters of the forward scattering signal ${\hat{t}}_{cont}$ and $\hat{V}$:(20)$${\hat{t}}_{cont}=t_{cont,i}\;\mathrm{and}\;\hat{V}=V_{j},\;\mathrm{if}\;\underset{i,j}{\max}Z_{i,j,T}\geq H$$

In the case of pulsar FSR, the direct and forward scattering signals are quasi-deterministic signals and are mathematically described as (12) and (13), respectively. If in the logarithm of the likelihood ratio (18) the signal $s_{direct}\left(t\right)$ is replaced by the expression (12) and the signal $s_{target}\left(t\right)$ is replaced by the expression (13), then the logarithm of the likelihood ratio in the multichannel algorithm for detection and estimation of parameters of the forward scattering signal from an airplane in each (i, j) channel, (i=1…M and j=1…N), has the form:(21)$$Z_{i,j,T}\left({\hat{t}}_{cont,i},{\hat{V}}_{j}\right)=\frac{2B}{N_{0}}\int_{0}^{T}p\left(t\right)\cos[2\pi (f_{0}+\frac{2{\hat{V}}_{j}}{\lambda })t]\sum_{k=1}^{\frac{{\hat{t}}_{cont,i}}{P}}\delta \left(t-kP\right)y_{0,i}\left(t\right)dt$$
(22)$$y_{0,i}\left(t\right)=y\left(t\right)-Ap\left(t\right)\cos\left(2\pi f_{0}t\right)\sum_{k=1}^{\frac{{\hat{t}}_{cont,i}}{P}}\delta \left(t-kP\right)\;\mathrm{and}\;T={\hat{t}}_{cont,i}$$

In the case of solar or lunar FSR, the direct and forward scattering signals from an airplane are mathematically described as Equations (14) and (15), respectively. After substituting $s_{direct}\left(t\right)$ and $s_{target}\left(t\right)$ with expressions Equations (14) and (15), the logarithm of the likelihood ratio takes the form:(23)$$Z_{i,j,T}\left({\hat{t}}_{cont,i},{\hat{V}}_{j}\right)=\frac{2}{N_{0}}\int_{0}^{T}D\left(t\right)\cos[2\pi (f_{0}+\frac{2{\hat{V}}_{j}}{\lambda })t]y_{0,i}\left(t\right)dt$$
(24)$$y_{0,i}\left(t\right)=y\left(t\right)-C\left(t\right)\cos\left(2\pi f_{0}t\right)\;\mathrm{and}\;T={\hat{t}}_{cont,i}$$

In this article, the proposed suboptimal algorithm for joint detection/estimation in the FSR system is described in the most general form from the point of view of the statistical radar theory, and the algorithm structure is presented. The structure is valid for various very weak cosmic signals, pulsed or continuous. Signals with unknown parameters—duration and speed—are detected. The estimation of the unknown parameters can be performed in one receiver, but it will take more time. We have proposed a more expensive, but real-time option, the use (multichannel measurement) of several receivers, each of which is set to certain values of the signal parameters—duration and speed.

Since the proposed structure describes both cases of signals (a series of pulse signals from pulsars and continuous signals from the Sun and the Moon), we have not gone into details and have considered that this is sufficient to reduce the description of this structure.

The structure of the detection algorithm described in the article can be realized with different approaches—correlation or filter. Each correlation or filter receiver is set to a specific signal length and speed. Before the detection in the receiving mixture of signals the direct signal is compensated in advance. Therefore, in this set of receivers matched to the re-radiated signal from the target, the strongest signal will appear at the output of one of them. This strongest signal will be compared to a pre-selected detection threshold. If the signal exceeds the threshold, the signal detection will be indicated as well as estimates of the signal length and speed (at which the receiver is set).

## 5. Power Budget of FSR Systems

*Pulsar FSR.* One of the very important parameters of pulsars, which are usually given in each pulsar database, is the average power spectral flux density $S_{av}$. This is the average power in watts transmitted by a pulsar per square meter per hertz. The peak power spectral flux density (S) transmitted by a pulsar can be evaluated using the basic pulsar parameters:(25)$$S=S_{av}P/W$$

The parameter W denotes the pulse width of the pulsar radio emission. The peak power of a single impulse at the antenna output can be evaluated as:(26)$$P_{imp}=S_{av}PA_{eff}\Delta f/W$$

According to [10,11,18,19], the power of the pulse sequence of $N_{p}$ pulses of the direct signal is:(27)$$P_{direct}=P_{imp}N_{p}=S_{av}PA_{eff}\Delta fN_{p}/W$$

The effective area of the receiver radio antenna is given by:(28)$$A_{eff}=2k_{B}G$$
where G is the receiver antenna gain (in units of K/Jy) and $k_{B}$ is the Boltzman constant (1.38 × 10^−23^ W/Hz/K). The power of the receiver noise is evaluated using the receiver system temperature $T_{sys}$ as:(29)$$P_{n}=k_{B}T_{sys}\Delta f$$

Taking into account Equations (27)–(29) and receiver gain q from Equation (8), the Signal-to-Noise Ratio (SNR) of the direct signal before compensation can be determined by:(30)$$SNR_{direct}=\frac{P_{direct}}{P_{n}}=\frac{2S_{av}GP}{T_{sys}W}qN_{p}$$

According to Equation (4), the SNR of the forward scattering signal from an airplane that has the silhouette area *Q* and is located at the distance *R* can be expressed by:(31)$$SNR_{det}=SNR_{direct}Q^{2}/\left(\lambda^{2}R^{2}\right)$$

Using the concept of Radar Cross Section (RCS), denoted as σ, Equation (31) can be rewritten as:(32)$$SNR_{det}=SNR_{direct}\frac{\sigma }{4\pi R^{2}}$$
where the airplane shadow RCS (σ) is expressed as:(33)$$\sigma =4\pi Q^{2}/\lambda^{2}$$

In case of airplane detection, when the airplane crosses the baseline “pulsar-receiver” at an almost right angle, we assume that the silhouette of the airplane is approximately equal to a rectangle with a length l and width h. Then Equation (29) takes the form:(34)$$\sigma =4\pi {\left(hl\right)}^{2}/\lambda^{2}$$

*Solar/lunar FSR.* The solar radiation spectrum covers all frequencies from the radio diapason to the optical one. In the optical diapason, the solar radiation flux is quite stable, with very small deviations over the solar cycle. However, in the radio frequency diapason, the solar radiance is different from that in the optical range. For that reason, daily observations of the Sun’s radio emission at a wide visible radio frequency diapason are conducted by different radio telescopes in the world. The solar observations (*S*) are presented as the power spectral flux density in solar flux units, where one solar flux unit (SFU) is 10^−22^ Wm^−2^Hz^−1^ or 10^4^ Jy. The radio emission of the Moon is purely thermal. The Moon re-emits the energy of solar radiation incident on it. At frequencies from the range 400 ÷ 1400 MHz, the brightness temperature $\left(T_{B}\right)$ of the Moon is about 240 K. This temperature can be used to calculate the power spectral flux density of the lunar radio emission (S). This is the power in watts transmitted by a transmitter (Moon) per square meter per hertz. Based on the Rayleigh–Jeans approximation for blackbody radiation at radio frequencies, the power spectral flux density can be approximately expressed as:(35)$$S=2k_{B}T_{B}\pi {\left(\frac{\alpha \pi }{180}\right)}^{2}\lambda^{-2}$$
where α is the angular radius of the Moon in degrees (α=0.25 deg).

The power of the direct signal received by the antenna with the effective area $A_{eff}$ is:(36)$$P_{solar}=SA_{eff}\Delta f$$

Taking into account Equations (26), (8) and (29), the SNR of the direct signal on the receiver output can be expressed by:(37)$$SNR_{direct}=\frac{P_{solar}}{P_{n}}=\frac{2SG}{T_{sys}}q$$

In analogy with Equation (32), the SNR of the forward scattering signal received from an airplane with the RCS σ located at the distance *R* can be expressed by Equation (32). In the case of lunar FSR the parameter S must be calculated assuming $T_{B}=240\;K$ in Equation (35) and $P_{solar}\;$in Equation (36), Equation (37) must be replaced by $P_{lunar}$. The expressions Equations (30), (32) and (37) are used for calculation, respectively, of $SNR_{direct}$ and $SNR_{det}$ in the Results section.

## 6. Probability Characteristics

In [19,20,21,22], calculations of the Signal-to-Noise Ratio depending on the distance to the aircraft were proposed, but they only show the potential use of FSR with cosmic radio emitters for detecting airplanes, asteroids, and meteorites. The results described in these articles are not directly related to the chosen probabilities of correct detection and false alarm, as well as to the problems of direct signal compensation.

This section proposes a methodology for evaluating the effectiveness of the proposed algorithm for joint compensation, detection and parameter estimation by determining the probability of correct detection and the maximum detection range. To simplify the problem of determining the probabilistic characteristics, we will first assume that the direct signal is fully compensated in all channels and that before calculating the logarithm of the likelihood ratio in the channels, there is only a mixture of the useful forward scattering signal and the receiver noise.

Only in this case can it be considered that statistics Equation (20) and statistics Equation (22) are the decision rules for detecting a deterministic signal against the background of white noise of the receiver and that known expressions can be used to calculate the probability of false alarm $\left(P_{fa}\right)$ and correct detection $\left(P_{D}\right)$ [3].
(38)$$P_{f\alpha }=1-\Phi \left(H\right)$$
(39)$$P_{D}=1-\Phi \left(H-SNR_{det}\right)$$
where Φ(.) is the probability integral. The threshold *H* is defined as the root of Equation (38) for a given false alarm probability level. The parameter $SNR_{det}$ in Equation (39) is calculated in the previous section for pulsar, solar and lunar FSRs depending on the frequency of emission, the size of airplanes, which are crossing the baseline “transmitter-receiver” at the right angle and also their distance [23,24,25].

Equation (39) can only be used when the direct signal is fully compensated.

In the case of incomplete compensation of the direct signal, the noise at the detector input is the sum of the white noise of the receiver and the residual direct signal after its compensation. In that case of incomplete compensation the SNR at the detector input $\left(SNR_{det}^{*}\right)$ is determined by:(40)$$SNR_{det}^{*}=\frac{P_{shadow}}{P_{direct}^{*}+P_{n}}=SNR_{det}\frac{1}{1+SNR_{direct}^{*}}$$

In Equation (40) the parameters $P_{direct}^{*}$ and $SNR_{direct}^{*}$ are, respectively, the power and SNR of the direct signal after compensation and $P_{n}$ is the power of the receiver noise.

Equation (40) in decibels can be written as:(41)$$SNR_{det}^{*}\left[\mathrm{dB}\right]\approx SNR_{det}\left[\mathrm{dB}\right]-SNR_{direct}^{*}\left[\mathrm{dB}\right]$$

The efficiency factor of the compensator $\left(I_{comp}\right)$ is usually defined as:(42)$$I_{comp}\left[\mathrm{dB}\right]=SNR_{direct}\left[\mathrm{dB}\right]-SNR_{direct}^{*}\left[\mathrm{dB}\right]$$

The parameter $SNR_{direct}$ in Equation (42) is calculated in the previous section for pulsar, solar and lunar FSRs depending on the frequency and intensity of emission, the receiver parameters, the antenna parameters and the contact time, which depends on the distance to the airplane.

It follows from Equation (42) that the $SNR_{direct}^{*}$ can be estimated as:(43)$$SNR_{direct}^{*}\left[\mathrm{dB}\right]=SNR_{direct}\left[\mathrm{dB}\right]-I_{comp}\left[\mathrm{dB}\right]$$

Therefore, in the case of the incomplete compensated direct signal, the probability of correct detection can be evaluated using the expression Equation (39) by replacing $SNR_{det}$ with $SNR_{det}^{*}$.

If the efficiency of the direct signal compensation exceeds the SNR value of the direct signal in Equation (43), i.e., $I_{comp}\left[\mathrm{dB}\right]>SNR_{direct}\left[\mathrm{dB}\right]$, then at the compensator output we have $SNR_{direct}^{*}\left[\mathrm{dB}\right]=0$ and $SNR_{det}^{*}\left[\mathrm{dB}\right]=SNR_{det}\left[\mathrm{dB}\right]$ in Equation (41).

It can be seen that the probability of correct detection of the proposed multichannel algorithm for joint detection and estimation of parameters with compensation of the direct signal (see Equations (17) and (18)) depends both on the selected detection algorithm and on the selected compensator and its efficiency.

## 7. Results

In this section, we have calculated the values of SNR for the direct $\left(SNR_{direct}\right)$ and the forward scattering signal from the aircraft $\left(SNR_{det}\right)$ depending on the distance to the aircraft (R).

The calculation is done for three different cosmic sources of radio signals (pulsar, Sun and Moon), as well as for three types of airplanes. Based on two graphs $\left(SNR_{direct}=f\left(R\right)\right)$ and $\left(SNR_{det}=f\left(R\right)\right)$, the maximum distance of correct detection $\left(R_{max,det}\right)$ can be graphically estimated. The distance $R_{max,det}$ is that at which two conditions are met: (1)—there is full compensation of the direct signal, i.e., $SNR_{direct}^{*}\left[\mathrm{dB}\right]=0$; (2)—with the value of $SNR_{det}$ the set level of probability for correct detection is achieved.

In calculations of SNR, the antenna with the diameter 25 m was used for the case of a pulsar and lunar FSR, and the antenna diameter of 1.3 m—for a solar FSR. The system temperature is assumed to be 150 K; the receiver frequency bandwidth is 100 MHz. All three types of FSR use the same receiver with high gain quality, and the same signal processing. In the calculation of SNR in a solar FSR, the density observation data were used (frequency 1415 MHz, spectral flux density 45 SUF) (prepared by the U.S. Dept. of Commerce, NOAA, Space Weather Prediction Center, 23 July 2018). The spectral flux density of lunar radiation was calculated using Equation (35) in the assumption that the brightness temperature of the Moon is 240 K.

The parameters of pulsar B0833-45 were taken from the EPN database [10]: the spectral flux density at a frequency of 1400 MHz is $S_{av}=1.1\;\mathrm{Jy}$; pulse width after epoch-folding is W=0.0021 s; repetition period is P=0.089328 s.

Three types of airplanes with different dimensions were selected for study, small—with the dimensions (h=2 m, l=10 m), medium—with the dimensions (h=3 m, l=20 m), and large—with the dimensions (h=4 m, l=30 m). These types of airplanes are assumed to have the following velocities: 100 m/s—for a pulsar and lunar FSR, and 200 m/s—for a solar FSR.

The SNR values at the detector input $\left(SNR_{det}\right)$ calculated for the three types of radiation (pulsar, Moon and Sun) are plotted in Figure 4, Figure 5 and Figure 6. The minimal distance is calculated as $R_{min}=Q/\lambda$ where Q is the airplane’s silhouette area.

From the graphs in Figure 7, Figure 8 and Figure 9 it follows that if we provide the efficiency of the compensator ($I_{comp}\left[\mathrm{dB}\right]$), respectively, equal to 37, 48 and 46 dB, then at the detector input we have the fully compensated direct signal (i.e., $SNR_{direct}^{*}\left[\mathrm{dB}\right]=0$), for all three FSR (pulsar, solar, lunar).

We can then use the graphs in Figure 4, Figure 5 and Figure 6 to estimate the maximum distance of correct detection under the condition (2), i.e., have the necessary $SNR_{det}=6\;\mathrm{dB}$ at the detector input in order to guarantee a probability of correct detection above 0.98 with a false alarm probability of 10^−4^ as shown in Figure 10.

Using the results in Figure 4, obtained for a pulsar FSR, it can be seen that small airplanes (h=1 m, l=2 m) cannot be detected by a pulsar FSR. In the case of the other airplanes (medium airplanes and large airplanes) the maximal distance of correct detection is as follows: 5 km—for medium airplanes (h=1 m, l=3 m) and 22 km—for large airplanes (h=3 m, l=5 m). The results in Figure 7 obtained for a solar FSR show that the maximal distance of correct detection is as follows: 7 km—for small airplanes, 24 km—for medium airplanes and 52 km—for large airplanes.

In the case of a lunar FSR (Figure 5), the maximal distance of correct detection is as follows: 5.7 km—for small airplanes, 24 km—for medium airplanes and 50 km—for large airplanes.

It should be noted that the pulsar FSR and the lunar FSR use a medium-sized antenna with a diameter of 25 m. To accumulate the SNR of the forward scattering signal sufficient to detect the aircraft, in this case, lower aircraft speeds are assumed (during takeoff and landing), which do not exceed 100 m/s.

However, due to the strong radiation flux from the Sun, the solar FSR uses a smaller antenna with a diameter of 1.3 m. For the same reason, long distances are assumed at the higher values of the airplane velocity (200 m/s).

It should be noted that the presented results do not take into account the real losses of signals during their reception and processing in the FSR.

The specificity of the FSRs using the sources cosmic emission is that they represent slow-moving barriers directed towards the cosmos. When the Sun, Moon and pulsar are located high above the horizon, the observation distance is minimal, because the planes do not fly at altitudes greater than 12 km. However, the time of observation in that case is long (hours). When the Sun, Moon and pulsar are very low above the horizon, the baseline of FSR is located close to the ground and, therefore, the time of observation is short (tens of minutes) and the distance of observation is large. In these specific situations of observation, it is possible to have conditions for the FS effect appearing in FSR when the airplanes are crossing the baseline at the approximately right angle. Only in this case the SNR values, obtained in this article for the different aircrafts, can be considered.

A stand-alone application for aircraft surveillance is not expected from these mobile FSR systems with cosmic sources of radio emission. Like classic radars with a circular survey of the airspace, these FSRs observe the airspace during the day. These properties can be very useful in solving many other tasks, such as ecology, meteorology and others.

They, like other passive radar systems and PCLs, can improve protection, surveillance and air traffic control in complex radar systems. These FSRs can complement these systems in conditions of particularly low-flying and small-sized objects in high city buildings or mountain ranges, and in environments with radio interference. The development of high-quality passive receiving systems, their microminiaturization and reduction in price are the ways of developing similar FSR systems using other cosmic sources of radio emission.

## 8. Conclusions

The article considers the problem of joint detection and estimation of the parameters of the forward scattering signal from an airplane with compensation of the direct signal from cosmic sources of radio emission (pulsar, Sun, Moon) for use in pulsar, solar and lunar FSRs. To solve this problem, this article proposes a new modified suboptimal multichannel algorithm for joint detection and estimation of the parameters of a quasi-deterministic forward scattering signal from an airplane, with compensation of a powerful direct quasi-deterministic signal from the cosmic source (Sun/Moon/pulsar) against the background of white Gaussian noise of the receiver. The unknown estimated parameters of the quasi-deterministic forward scattering signal from the airplane are the signal duration and the Doppler velocity of the aircraft. A technique is proposed for assessing the input SNR of direct and forward scattering signals, as well as for assessing the maximum detection distance, taking into account the probability of correct detection and the probability of false alarm.

The proposed approach for estimating the maximum distance of detection makes it possible to form the requirements for the efficiency of the direct signal compensator. The results obtained for the maximum detection ranges of three types of airplanes (small, medium and large) using FSRs with cosmic radio emitters (pulsar, Sun and Moon) do not take into account the actual signal processing losses in the FSR. However, they show the potential for implementing FSRs using various cosmic emitters (pulsars, Sun and Moon) in sophisticated radar systems. Upon specific time situations, conditions arise for the formation of the FS effect between the transmitter and the radar, when crossing the flying airplanes, the baseline between the receiver and the transmitter. Due to the FS effect, these FSRs can improve the efficiency of radar systems in air traffic protection and surveillance, complementing them in situations with particularly low-flying and small objects, in tall urban buildings or mountain ranges, as well as in radio counteraction.

## Figures and Tables

**Figure 1 sensors-21-00465-f001:**
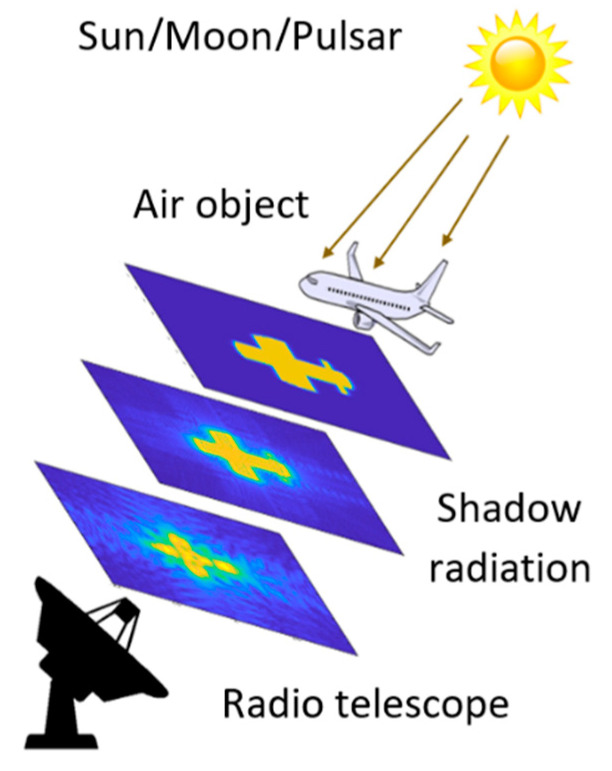
Topology of a Forward Scatter Radar (FSR) system using cosmic radio emission.

**Figure 2 sensors-21-00465-f002:**
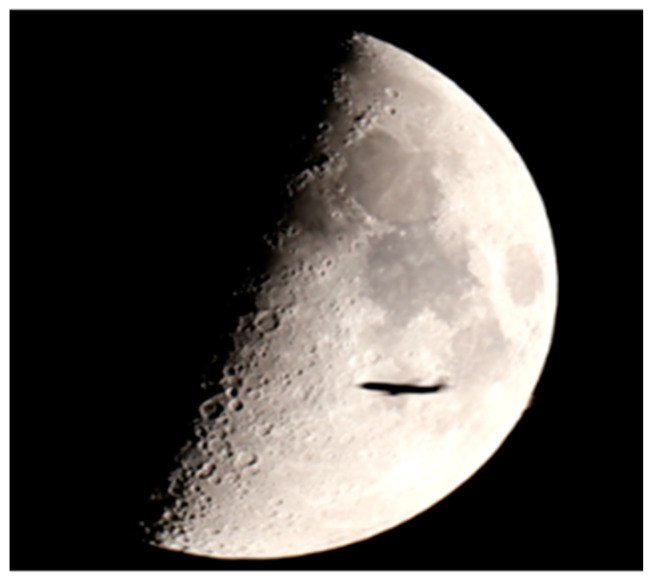
The airplane shadow against a lunar background.

**Figure 3 sensors-21-00465-f003:**
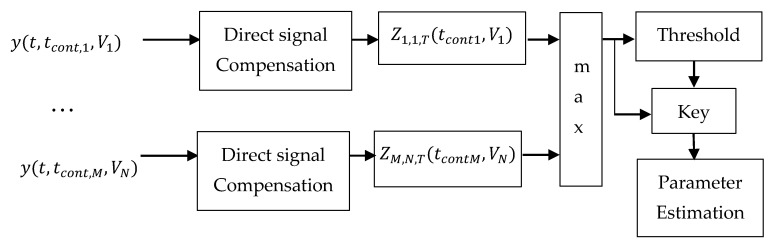
Algorithm for detection of the forward scattering signal from an airplane.

**Figure 4 sensors-21-00465-f004:**
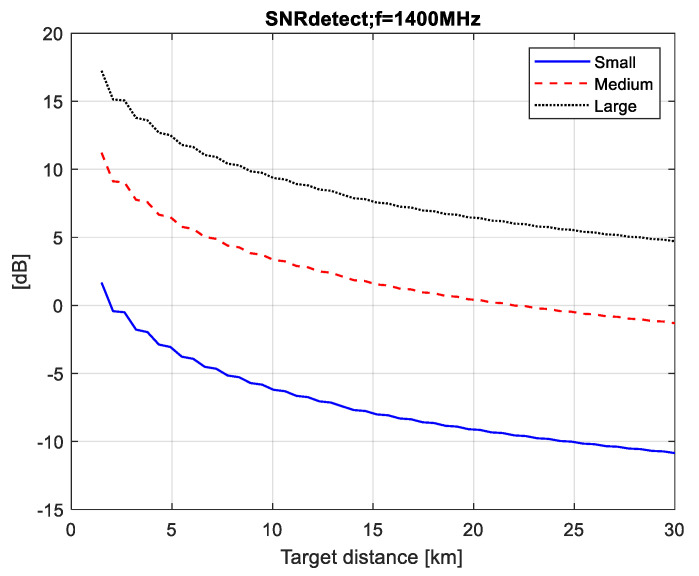
$SNR_{det}$ values for pulsar FSR.

**Figure 5 sensors-21-00465-f005:**
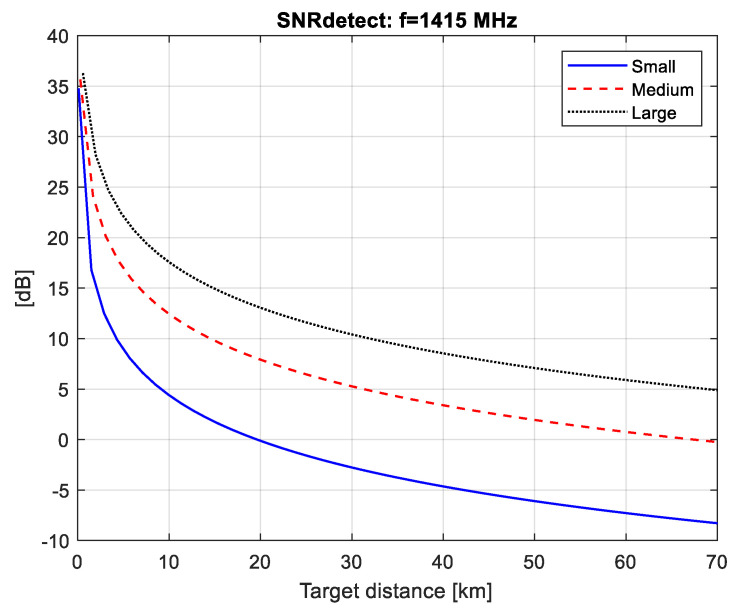
$SNR_{det}$ values for solar FSR.

**Figure 6 sensors-21-00465-f006:**
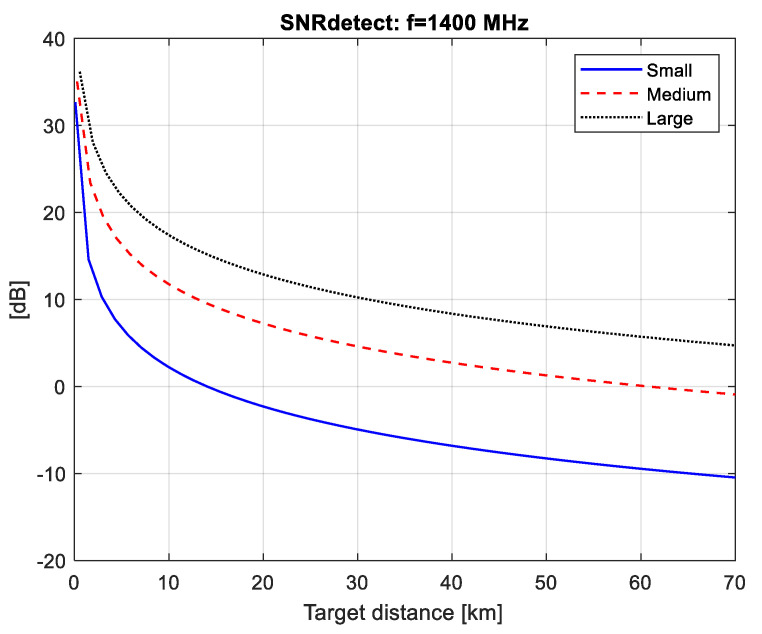
$SNR_{det}$ values for lunar FSR.

**Figure 7 sensors-21-00465-f007:**
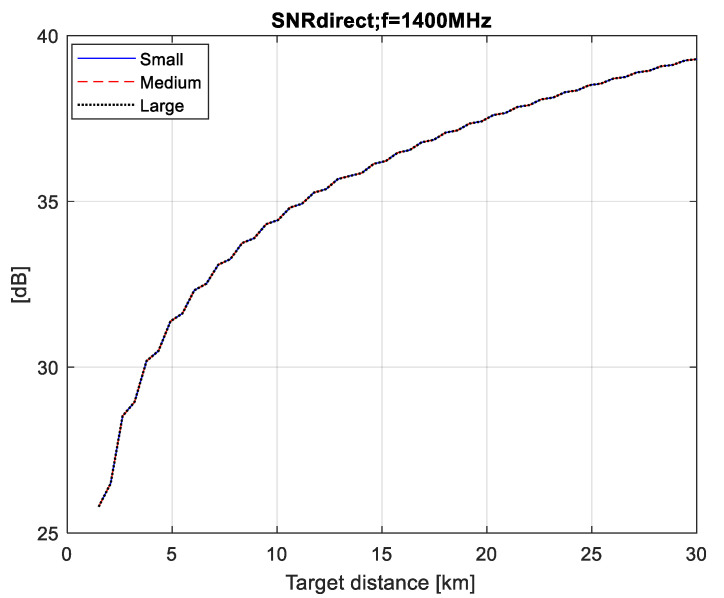
$SNR_{direct}$ values for pulsar FSR.

**Figure 8 sensors-21-00465-f008:**
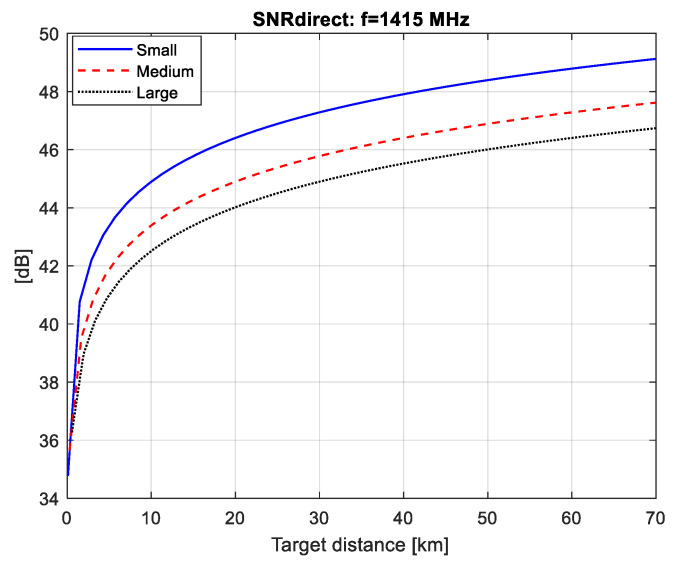
$SNR_{direct}$ values for solar FSR.

**Figure 9 sensors-21-00465-f009:**
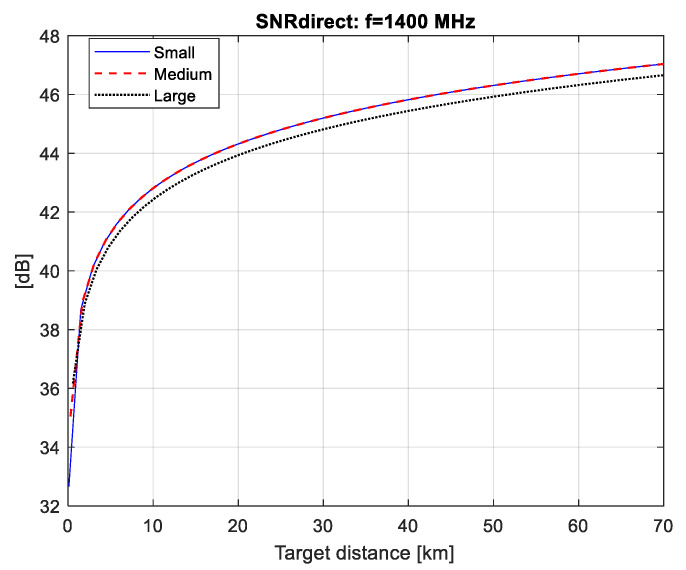
$SNR_{direct}$ values for lunar FSR.

**Figure 10 sensors-21-00465-f010:**
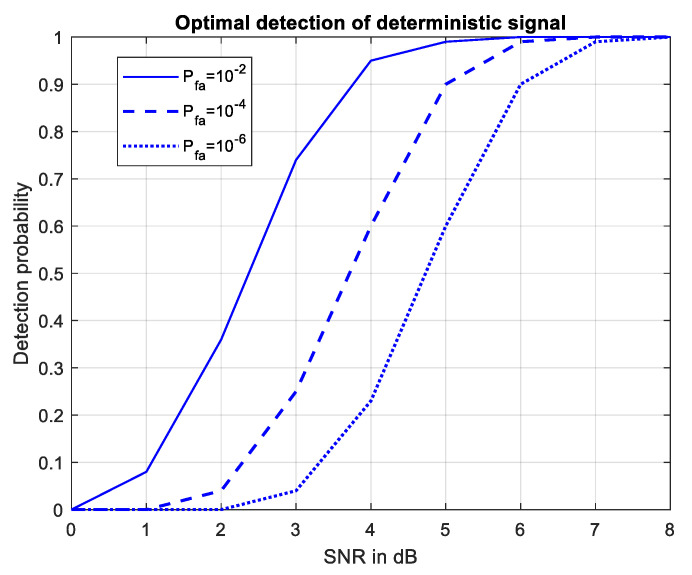
Probability of detection for various $P_{fa}$.

## Data Availability

The data presented in this study are available on request from the corresponding author.

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
