# Peer review of "FSR Systems for Detection of Air Objects Using Cosmic Radio Emissions"

_sensors, 2021, doi:10.3390/s21020465_

Round 1

Reviewer 1 Report

The paper presents a method to detect airplanes using the shadow signal from cosmic sources of radio emission with FSR systems. The work is interesting, however, the principles of the algorithms as well as the limitations of the approach need to be more clearer.

  1. According to the principles of the method, it seems that only the airplanes crossing the baseline of transmitter and receiver could be detected using the proposed approach. It will be a large limitation in practical applications of such an approach. Whether the range and direction of the airplanes can be estimated? The applicability of the proposed approach needs to be carefully analyzed.

  1. In the introduction, what does the ''unwanted objects'' mean?

  1. In section 4, what is the model of the multichannel processing? Does it use an array receivers? It is not clearly stated in the paper. In addition, a detailed detection algorithm should be given, rather than a ''structure'' of the algorithm.

Author Response

Authors' response to a review of an article

“FSR Systems for Detection of Air Objects using Cosmic Radio Emissions”

Reviewer 2 Report

  1. In Formula (3), it is proposed that R does not change with dq when the distance R is far. However, when the frequency is 1400Mhz, the wavelength is about 0.2m. Although the relative change of R on different dq is relatively small, but it still causes the phase change of each dq, so the change of R cannot be ignored directly. Therefor the approximation from formula (2) to formula (3) cannot be established.
  2. In formula (17), when the estimated values of the first and the second integrals are completely matched, only the correlation integral between target echo and noise exists, which should be the minimum value, so the maximum value proposed in Formula (20) is unreasonable. Please analyze and explain.
  3. The algorithm in this paper is carried out under the condition that the compensation is completely accurate, but the compensation error of direct signal cannot be eliminated completely. If this error exists, whether the accurate parameter estimation can be obtained.
  4. How to determine the log likelihood ratio threshold in Figure 3?
  5. When there are multiple targets, whether the target can be separated, if so, how to determine the duration tcont and how to correspond with the velocity V.
  6. After removing the direct signal, the target signal (shadow signal) is still weak. Under low signal-to-noise ratio (SNR), whether the target parameter estimation can be realized in Equation (18), whether there are requirements for detecting SNR, and if so, what are the requirements? Please give the theoretical analysis or simulation instructions.

Author Response

(The authors gave the same response as above.)

Reviewer 3 Report

General description

The concept of forward scattering radar and algorithm for target’s detection and parameter estimation based on Solar, Lunar and pulsar emission is in the focus of the present work. The probability characteristics and power budget for different emission sources are discussed. Recommendation to depress the stronger direct signal are also given.

The question is what is the role of the small delay of the scattered signal relative to the direct transmitter signal, which directly arrives at the receiver input in case the direct signal must be depressed.  Is there any meaning the distance between the target and space sources – pulsars, Solar, and Lunar.

The term shadow signal is not understandable. The shadow is the optical effect of obstructing the passing of the light or electromagnetic beam trough the object.  The object can be seen as a shadow based on the difference in the intensities of two zones, illuminated and not illuminated (shadow zone). In this case, the illuminated zone is the zone with a maximum intensity; the shadow zone is the zone with a zero intensity, no light, no signal, simple nothing.  The shadow does not have a gradient of the intensity, except at the edges of the object silhouette due to the diffraction.

There is a controversy in the terms “shadow radiation” and “shadow signal”. The shadow zone is a passive one, without emissions, a zone that can be seen on the background of the illuminated zone.

See attached file.

Author Response

(The authors gave the same response as above.)

Reviewer 4 Report

Original Submission

Recommendation

Major revision

Comments to Author:

Title:  

FSR Systems for Detection of Air Objects using Cosmic Radio Emissions

Overview and general recommendation.

This paper is demonstrating a system to detect air objects with cosmic radiations. These kinds of data are always welcomed to MDPI journals including RS, Sensors, Imaging etc. I really like the material: it is very advanced and interesting. The abstract is OK; but the Introduction is NOT: you should explain more the techniques, and the state of the art MUCH way better; pls rewrite it again. Honestly, I like the work: very good and efficient, perfect explanation. I think section 5 (NOT to mentioned that YOU have two section 5!!!) is not necessary: we all know how to reach to SNR or RCS… . The results section must be developed. Comparison with the other methods (Much experiments, and reliable comparisons must be presented), and the accuracy/precision must be addressed too. Conclusion is OK. English is fair.

I have a general question: why you do not use KALMAN filtering which seems much suitable for this kind of work? How about Bootstrapping parameters selections? Or other similar methods? 

Detailed comments:

line 56. Put a ref here.

lines 75-77. Rephrase please. I cannot understand.

line 96. Put a ref here.

line 97. Put more refs here. I think we need to convince the reader that the integral is the model that have been used before.

line 125. Put a ref here: I do not get why the Eq (9) holds.

line 128. Redundancy: we know Lambda is wavelength; do not repeat it here.

line 132-134. Can you put a graph (2D) showing all the parameters on it? I mean a signal (like Sin signal) showing the parameters mentioned in the paragraph.

line 139. Put a ref here.

line 157-159. Assuming…: How much this assumption is valid?

line 163. Put a ref for b<<1.

line 184. Explain why the noise is Gaussian.

line 194. Put a ref here.

Author Response

(The authors gave the same response as above.)

Round 2

Reviewer 1 Report

The manuscript has been revised according to the advice. Some explanations were made with respect to the concerns. It is recommended that more detailed analyses for applicability of the proposed approach be made in the future. 

Author Response

Dear Reviewer,

Thank you very much for putting efforts and time in the reviewing of our paper. Your comments and remarks were very important and really helped to improve our work.

Best regards,

The authors

Reviewer 2 Report

I think the author made necessary response and explanation to the comments, and have made corresponding improvement in the revised version. I agree with the author's response and believe that these responses will help readers better understand the content of the manuscript.

Author Response

(The authors gave the same response as above.)

Reviewer 3 Report

https://en.wikipedia.org/wiki/Diffraction#:~:text=Diffraction%20refers%20to%20various%20phenomena,shadow%20of%20the%20obstacle%2Faperture.

"Diffraction refers to various phenomena that occur when a wave encounters an obstacle or opening. It is defined as the bending of waves around the corners of an obstacle or through an aperture into the region of geometrical shadow of the obstacle/aperture."

Hence, the so-called shadow signal does not exist. There exist only the shadow of the obstacle equal to the shadow of the aperture according to Babinet principle. There exist a phenome diffraction, diffraction pattern or diffraction signal which cannot be called shadow signal.

There only exist a bending of waves around the corners of an obstacle not a hole. The wave front is not interrupted it is continuous and defines the diffractive signal.

The authors must to know that, the words profile, section, shape, side, contour, silhouette are synonymous. It means shape is equal to silhouette. However, the shape has to refer to the real geometry of the real object, which the waves interact with, whereas the silhouette may refer to object’s image which is not a real object.

Author Response

Dear Reviewer,

Thank you very much for putting efforts and time in the reviewing of our paper. Your comments and remarks were very important and really helped to improve our work. The adjustments made by us are described in the attached file.

Best regards,

The authors

Reviewer 4 Report

The authors have been responded almost all of the comments, and seems to me they are in good understating of the material; the paper is improved remarkably, and in this form, I am positive to accept that,

Good Luck!

Author Response

(The authors gave the same response as above.)
